# The Peak Plasma Concentration (Cmax)/Minimum Inhibitory Concentration (MIC) of bedaquiline and levofloxacin with special attention to the sputum conversion in the treatment of multidrug-resistant tuberculosis in Indonesia

Nina Mariana [1,2], Anggi Gayatri[3], Indah Suci Widyahening[4], Budiman Bela[5], Soedarsono Soedarsono[6], Iin Maemunah[7], Rosamarlina Rosamarlina[8], Vivi Setiawaty[2], Vithal Prasad Myneedu[9], Erni Juwita Nelwan[10], Purwantyastuti Ascobat[3]*

1 Doctoral program in Medical Sciences, Faculty of Medicine, Universitas Indonesia, Jakarta, Indonesia, 2 Directorate Of Human Resources-Education and Research, National Infectious Disease Center - Sulianti Saroso Hospital, Jakarta, Indonesia, 3 Department of Pharmacology and Therapeutics, Faculty of Medicine, Universitas Indonesia, Jakarta, Indonesia, 4 Department of Community Medicine, Faculty of Medicine, Universitas Indonesia, Jakarta, Indonesia, 5 Department of Microbiology, Faculty of Medicine, Universitas Indonesia, Jakarta, Indonesia, 6 Sub-Pulmonology Department of Internal Medicine, Faculty of Medicine, University of Hang Tuah, Surabaya, Indonesia, 7 Department of Microbiology at Pulmonary Hospital, Dr. M. Goenawan Partoidigdo Hospital, Bogor, Indonesia, 8 Departement of Pulmonology, National Infectious Disease Center - Sulianti Saroso Hospital, Jakarta, Indonesia, 9 Department of Microbiology at Nepalgunj Medical College Teaching Hospital (NGMC), Nepal, 10 Division of Tropical Medicine and Infectious Diseases, Department of Internal Medicine, Faculty of Medicine, Universitas Indonesia, Cipto Mangunkusumo Hospital, Jakarta, Indonesia

☯ These authors contributed equally to this work.
* purwanty2703@yahoo.com

## Abstract

### Background

Tuberculosis in Indonesia remains a serious public health concern, as the country has the third-largest number of multidrug-resistant tuberculosis (MDR-TB) patients in the world. Bedaquiline and levofloxacin are the primary drug regimen for MDR-TB treatment in Indonesia. This study aimed to investigate whether the Cmax/MIC of bedaquiline and levofloxacin differs between patients with sputum conversion and those without sputum conversion during the first four months of MDR-TB treatment in Indonesia.

### Methods

A cohort study was performed in adult patients (18–65 years old) treated with the 18–24-month oral regimen. Patients were excluded if they were pregnant or HIV positive, with uncontrolled diabetes, or had any of the following severe conditions: cancer, digestive, cardiovascular system disorders, hepatic, or renal problems. Two blood samples were collected to measure the plasma concentrations of bedaquiline

**Data availability statement:** Data Availability: All relevant data are within the manuscript and its supporting Information files.

**Funding:** This study was funded by [Universitas Indonesia http://dx.doi.org/10.13039/501100006378 NKB_292 Prof Purwantyastuti Ascobat]. We have no conflicts of interest to disclose. The funders had no role in study design, data collection and analysis, decision to publish, or preparation of the manuscript.

**Competing interests:** The authors have declared that no competing interests exist.

and levofloxacin using liquid chromatography-tandem mass spectrometry (LC-MS/MS). *Mycobacterium tuberculosis* (Mtb) isolates from patients' sputum were used to determine the MIC of individual drugs using liquid growth media (MGIT).

## Results

Among the 74 patients enrolled, 16 dropped out during the four-month follow-up period. Blood samples were successfully obtained 1–2 hours after drug administration from 48 patients and 4–6 hours after drug administration from 32 patients, which were used for pharmacokinetic analysis. Sputum conversion was detected in 84.5% of patients during four months of the MDR-TB treatment. The mean $C_{max}$/MIC ratio of bedaquiline was higher in the sputum conversion group compared with the non-conversion group (9.10 vs. 1.65, respectively). Meanwhile, a small difference in the $C_{max}$/MIC ratio of levofloxacin was observed between the sputum conversion group and the non-conversion group; it was not significant (45.64 vs. 41.72, respectively; p = 0.941).

## Conclusions

$C_{max}$/MIC of bedaquiline was higher in MDR-TB patients with sputum conversion compared to those without conversion within the first four months of treatment, suggesting a potential relationship between $C_{max}$/MIC of bedaquiline and sputum conversion, which was not seen in the levofloxacin cases. However, the application of clinical practice should be carefully considered and supported by further study in various settings.

## Introduction

The World Health Organization (WHO) declared tuberculosis (TB) a global emergency due to approximately 1.25 million TB deaths reported among cases in 2023 [1]. The highest prevalence of TB is in Southeast Asia (45%), followed by Africa (24%), the Pacific (17%), and other regions. TB treatment is increasingly complicated by the rise of drug-resistant TB cases, which are escalating rapidly each year. Indonesia ranks as the second country with the highest TB burden and the third highest number of MDR-TB cases [1,2]. As of 2024, the success rate of MDR-TB treatment in Indonesia stands at 58%, significantly below the global target level. Data from 2015 to 2022 indicated that the average percentage of deaths and lost-to-follow-up cases among 1053 MDR-TB patients in Indonesia was 12% and 27%, respectively [3]. MDR-TB treatment generally requires 18–24 months and is often linked to frequent and severe side effects [4]. The low success rate, interindividual variability in the pharmacokinetics of anti-TB drugs, and the lack of minimum inhibitory concentration (MIC) data in some countries, including Indonesia, contribute to an unclear understanding of treatment challenges. This lack of clarity highlighted the need to improve the effectiveness of treatment to prevent the evolution of resistance [5–7]. A comprehensive understanding of appropriate drug treatments is essential in Indonesia.

Currently, there are three types of regimens for the treatment of MDR-TB, one of which is the 18–24 months all-oral regimen. Two key drugs (group A) in this regimen are bedaquiline and levofloxacin. Bedaquiline and levofloxacin are antibiotics that show a concentration-dependent pattern of activity [6,8]. The Cmax/MIC ratio integrates pharmacokinetic (Cmax) and pharmacodynamic (MIC) for drug use optimization that can inhibit bacterial growth, improve patient outcomes, and reduce the risk of resistance development. It is important to know the plasma concentrations of bedaquiline and levofloxacin compared to their MICs for *Mycobacterium tuberculosis* (Mtb) in areas with a high incidence of drug-resistant TB [9–11].

The MIC is the lowest concentration of an anti-TB drug required to inhibit the visible growth of Mtb in vitro. MIC values may also be higher or lower than in other countries and regions due to resistance patterns. The susceptible category for both levofloxacin and bedaquiline was determined to have MIC ≤ 1 mg/L using MGIT960 [12,13]. The MIC value may contribute to the variable rates of pharmacokinetic/pharmacodynamic that are considered effective for TB treatment [14,15].

The plasma concentrations of anti-TB drugs, as well as their concentrations at the infection site (in the lungs for pulmonary TB), should be considered to predict clinical outcome [16,17]. The drug concentrations in tissues are difficult to measure, and plasma concentrations are typically used as a surrogate [16]. Therefore, to assess clinical outcomes, it is important to determine the plasma concentrations of bedaquiline and levofloxacin in Indonesian patients receiving the dosages according to WHO Guidelines, as well as to understand the MIC in the area. The effectiveness of MDR TB treatment is usually monitored through sputum culture conversion, which serves as a key clinical outcome indicator [18]. As part of a preliminary study on plasma concentrations of bedaquiline. The present study aimed to investigate whether the Cmax/MIC of bedaquiline and levofloxacin differs between patients with sputum conversion and those without sputum conversion during the first four months of MDR-TB treatment in Indonesia.

## Methods

### Study design and setting

A cohort study was conducted in three referral hospitals in Bogor and Jakarta, Indonesia, from December 2023 to December 2024. The referral hospitals in Jakarta were the National Infectious Disease Center, Sulianti Saroso Hospital, and Pasar Rebo Regional General Hospital. A referral hospital in Bogor was Pulmonary Hospital Dr. M. Goenawan Partowidigdo. The estimated sample size was calculated for each bedaquiline Cmax/MIC and levofloxacin Cmax/MIC variable in the sputum conversion group and non-conversion group, using hypothesis tests that compared the means of two independent populations. MedCalc Software Ltd was used, and 57 samples were required. Additional sample size calculation output (screenshots and MedCalc parameters) is available in S1 File. Patients were enrolled using consecutive sampling procedures.

### Patients and criteria

This study included adult MDR-TB outpatients aged 18–65 years treated with the 18–24 month all-oral regimen for MDR-TB (bedaquiline, levofloxacin, linezolid, clofazimine, and cycloserine) for at least seven days and before four weeks of treatment. Patients were excluded if they were pregnant or HIV positive, with uncontrolled diabetes, or had any of the following severe conditions: cancer, digestive, cardiovascular system disorders, hepatic, or renal problems. If patients missed their medication three times within the four-month study period, they were considered non-compliant for any reason and dropped from this study. The clinical outcome in terms of sputum conversion was evaluated within four months of treatment during the study period. Covering the possibility of dropout, 74 adult patients were enrolled in the study. Seventy-four patients were new patients who were treated with the MDR-TB regimen in December 2023 and September 2024 at three hospitals. All patients received five drugs, including bedaquiline (400 mg daily for the first two weeks followed by 200 mg three times per week), levofloxacin (750 or 1000 mg/day), linezolid (300 or 600 mg/day), clofazimine (100 mg/day), and cycloserine (250 or 750 mg/day). During the first month of treatment, most patients took their medication in

the morning at the hospital, and patients routinely visited the hospital's outpatient department every day as part of the MDR-TB program.

## Data collection and measurements

The patient's demographic data were collected, including baseline laboratory data, baseline chest radiograph, and electro-cardiogram, which were obtained from the electronic medical record.

At the baseline treatment, resistant Mtb isolates from patients' sputum were examined for the MIC of bedaquiline and levofloxacin. The material of pure drug sigma-Aldrich 28266-IG levofloxacin and the pure drug bedaquiline was accessed via https://www.beiresources.org/, which were used for the determination of individual drugs' MIC by liquid growth media (MGIT) at the Jakarta Centre for Biomedical and Health Genomics (BBinomika) laboratory, Indonesia. Serial dilutions of bedaquiline and levofloxacin were 0.06, 0.12, 0.25, 0.50, and 1 mg/L prepared in MGIT tubes. The determination of the estimated concentration range was based on several previous studies, which showed that this range could represent the MIC value of the MDR TB strain [13,19,20].

In the fourth week of treatment, two blood samples were taken on the same day to detect both the plasma concen-trations of bedaquiline and levofloxacin. Previous studies had shown that steady-state concentrations were achieved by the fourth week of treatment for bedaquiline and levofloxacin. Levofloxacin concentration reached peak concentration (Cmax) in plasma at 1–2 hours, while the bedaquiline concentration reached Cmax in plasma at 4–6 hours after drug administration [21,22]. Six milliliters of venous blood were collected at each time point (1–2 hours and at 4–6 hours). The samples were cen-trifuged at 3000 rpm for 5 minutes within 2 hours of collection. The separated plasma samples were stored in a –80°C freezer.

Liquid chromatography-tandem mass spectrometry (LC-MS/MS) was developed and validated to determine plasma con-centrations of bedaquiline and levofloxacin. The LC-MS/MS system utilized Xevo TQD, with positive mode multiple-reaction monitoring (MRM) using an electrospray ion source (ESI). The instrument consisted of a Waters Acquity UPLC® HSS T3 C18 (1.8 mm) 2.1 x 50 mm, and an analytical column at 40°C. The mobile phase consists of 0.1% formic acid in water and aceto-nitrile, with a gradient elution flow rate of 0.3 mL/minute. MRM modes used mass transition 555.20/58.11 m/z for Bedaquiline and 362.34/318.11 for levofloxacin, and 384.20/253.12 for quetiapine as internal standard. The lower limit of levofloxacin detection in plasma was 0.04 mg/L, and for bedaquiline was 0.004 mg/L. The accuracy and precision of this assay met the standards set for bioanalytical method validation. The laboratory was located at Pharmacometrics, Jakarta, Indonesia.

The Cmax/MIC ratio was subsequently calculated for each patient by dividing Cmax by the MIC value. Sputum conver-sion was assessed after four months during the early phase of treatment. Sputum conversion was defined as at least two consecutive negative acid-fast bacilli (AFB) smear microscopy and cultures.

## Data management and statistical analysis

The collected data were compiled and analyzed using SPSS statistical software version 24.0. Quantitative data with a normal distribution are presented as the mean (standard deviation), and non-normally distributed data are presented as the median (maximum-minimum). A comparative analysis was conducted between the conversion and non-conversion groups. An independent samples t-test was used for continuous data with a normal distribution. A non-parametric Mann–Whitney U test was used when data were not normally distributed. The chi-square test or Fisher's exact test was used to compare categorical data between the two groups. Statistical significance was set at $p < 0.05$.

## Ethical approval

This study was approved by the Health Research Ethics Committee of the Faculty of Medicine, Universitas Indonesia (KET-1497/UN2).F1/ETIK/PPM.00.02/2023). Written informed consent was obtained from all patients, and all clinical investigations were conducted by the principles of the Declaration of Helsinki.

## Results

The characteristics of the subjects in this study are shown in Table 1. During the four months of treatment, out of 74 patients, 16 dropped out, and 58 patients completed the analysis. Information regarding patients who dropped out is explained in the last paragraph of this section. Fig 1 shows the patient's recruitment and follow-up flowchart. A total of 74 patients who met the inclusion criteria and did not meet the exclusion criteria were enrolled from new patients treated with MDR-TB regimens in December 2023 and September 2024 at three hospitals. Fifty-eight patients were successfully followed up for four months of treatment; however, 48 plasma samples were successfully analyzed for levofloxacin Cmax, and 32 plasma samples for bedaquiline.

At the end of the fourth month of treatment, the number of patients with sputum conversion was 49 (84.5%), while non-conversion was nine (15.5%).

Plasma samples of 50 patients (eight patients did not arrive as planned) were collected at 1–2 hours after drug administration, and 48 plasma samples were successfully analysed for the Cmax of levofloxacin (one of the samples was lysed, and one sample showed LLOQ). These plasma samples consisted of 41 samples belonging to the conversion groups and 7 samples to the non-conversion groups (Fig 1). Only 34 plasma samples were collected in the next 4–6 hours on the same day, of which only 32 samples could be analyzed (one sample lysis, one sample showed LLOQ), while 24 patients skipped their second blood sampling appointment due to illness, feeling unwell, conflict with their work activities, or did not show up on time with no information. Additional baseline characteristics of the group with missing bedaquiline or levofloxacin Cmax data and the group without missing Cmax data are provided in S2 Table.

Most patients were male in both the conversion and non-conversion groups. One hundred percent of patients in the non-conversion group had experienced previous TB treatment. All patients in both groups had lung lesions with moderate or far-advanced lesions. Table 1 shows the demographic distribution and clinical characteristics of the study population based on sputum conversion.

The MIC distributions of bedaquiline and levofloxacin are drawn in Fig 2. The MICs for bedaquiline and levofloxacin were obtained from 51 out of 58 resistant Mtb isolates from sputum (five isolates were not stored, and two did not grow). In this study, the MIC of bedaquiline showed that 41.2% of Mtb strains were equal to 0.25 mg/L; 35.5% of strains were under 0.12 mg/L, and 23.3% of strains were ≥ 0.50 mg/L. Meanwhile, regarding the MIC of levofloxacin, most (68.6%) of the Mtb strains in this study were equal to 0.25 mg/L (Fig 2). The MIC values for bedaquiline and levofloxacin, based on clinical outcome (sputum conversion or non-conversion), did not show significant differences (Table 2).

The mean Cmax of levofloxacin obtained from the conversion group was 11.36 (SD 5.69) mg/L, and the non-conversion group was 10.97 (6.56) mg/L, while the median of Cmax/MIC was 45.64 (17.14–206.23) mg/L in the conversion group, and 41.72 (23.06–86.68) mg/L in the non-conversion group. There was no significant difference between the Cmax and Cmax/MIC ratio of levofloxacin in the conversion and non-conversion groups.

The mean Cmax of bedaquiline obtained from the conversion group was 1.54 (SD 0.76) mg/L, and the non-conversion group was 0.70 (SD 0.44) mg/L. There was a significant difference (unpaired t-test, $p = 0.041$) between the Cmax of bedaquiline in the conversion and non-conversion group; similarly, there was a higher Cmax/MIC ratio of bedaquiline in the conversion compared with the non-conversion group (9.10 vs. 1.65; $p < 0.001$). Comparison of bedaquiline Cmax/MIC ratio and levofloxacin Cmax/MIC ratio with sputum conversion is shown in Tables 3 and 4. Summary statistics for the mean, median, variance, and normality of the Cmax, Cmax/MIC data in the groups are available in S3 Table and S4 Table.

The difference in Cmax/MIC based on the outcome (sputum conversion) is clearly shown in Fig 3. Fig 3 shows a summary of outcomes (conversion or non-conversion) based on levofloxacin, bedaquiline Cmax, and their MICs. Additional baseline characteristics of 30 patients who had Cmax/MIC bedaquiline data are provided in S5 Table.

Fig 4 shows the mean bedaquiline Cmax compared to the individual bedaquiline MIC value for Mtb, and the target Cmax/MIC ratio refers to the target Cmax/MIC ratio for antibiotics that show a concentration-dependent activity pattern

**Table 1. Demographic data and clinical characteristics of 58 MDR-TB patients at the start of treatment (baseline).**

| Characteristics | Conversion (four-month) (n = 49) | Non-conversion (four-month) (n = 9) | p value |
|---|---|---|---|
| Age (years), mean (SD) | 39 (12) | 35 (11) | 0.363[t] |
| Sex, n (%) | | | |
| Male | 32 (65.3) | 5 (55.6) | 0.710[f] |
| Female | 17 (34.7) | 4 (44.4) | |
| Contact with TB cases, n (%) | 8 (16.3) | 1 (11,1) | 1.000[f] |
| History of tobacco use, n (%) | 27 (55.1) | 5 (55.6) | 1.000[f] |
| History of alcohol use, n (%) | 10 (20.4) | 1 (11.1) | 1.000[f] |
| Previous TB treatment, n (%) | 31 (63.3) | 9 (100.0) | 0.045[f] |
| Comorbidity (well-controlled diabetes) | 2 (4.1) | 0 | 1.000[f] |
| Categorical BMI, n (%) | | | |
| Normal weight (18.5 to 24.9 kg m$^2$) | 18 (36.7) | 1 (11.1) | 0.130[f] |
| Underweight (< 18.5 kg m$^2$) | 31 (63.3) | 8 (88.9) | |
| Lung lesion, n (%) | | | |
| Moderate advanced | 6 (12.2) | 0 | 0.576[f] |
| Far advanced | 43 (87.8) | 9 (100.0) | |
| Laboratory results | | | |
| Hemoglobin (g/dL), mean (SD) | 11.9 (1.7) | 11.2 (1.9) | 0.273[t] |
| Hematocrit (10^6/µL), mean (SD) | 38.2 (5.9) | 36.2 (5.5) | 0.347[t] |
| Eritrosit (10^6/µL), mean (SD) | 4.7 (0.5) | 4.5 (0.7) | 0.407[t] |
| Leukosit (µL), mean (SD) | 9985 (3868) | 9595 (4482) | 0.787[t] |
| Trombosit (10^3/µL), mean (SD) | 378 (122) | 396 (140) | 0.693[t] |
| MCV (fL), median (min-max) | 82 (62–93) | 79 (71–84) | **0.031[m]** |
| MCH (pg), mean (SD) | 26 (2.8) | 24 (1.5) | 0.081[t] |
| MCHC (g/dL), median (min-max) | 32 (29–36) | 31 (29–33) | 0.150[m] |
| Neutrofil Limfosit Ratio, median (min-max) | 4.0 (1.4–14.5) | 6.5 (2.3–22.7) | 0.138[m] |
| AST (IU/L), median (min-max) | 19.0 (4–70) | 14.0 (11–28) | 0.053[m] |
| ALT (IU/L), median (min-max) | 17.0 (5–66) | 12.0 (7–20) | 0.083[m] |
| Creatinine (mg/dL), mean (SD) | 0.7 (0.2) | 0.6 (0.1) | 0.486[t] |
| Albumin (g/L), median (min-max) | 3.3 (2.0–4.9) | 3.1 (2.2–4.3) | 0.502[m] |
| Other drug use, n (%) | | | |
| Analgesics | 16 (32.7) | 1 (11.1) | 0.258[c] |
| Proton Pump Inhibitor (PPI) | 17 (34.7) | 3 (33.3) | 1.000[f] |
| Antiemetic | 5 (10.2) | 0 | 1.000[f] |
| Antifungal | 1 (02.0) | 2 (22.2) | 0.060[f] |
| Antiulcerant | 9 (18.4) | 1 (11.1) | 1.000[f] |
| Iron supplement | 4 (8.2) | 4 (44.4) | **0.015[f]** |

[f]Fisher's exact test; [t]unpaired t test; [m]Mann-Whitney U test; MCHC: Mean corpuscular hemoglobin concentration, MCH: Mean corpuscular hemoglobin, MCV: Mean corpuscular volume; AST: aspartate aminotransferase; ALT: alanine aminotransferase.

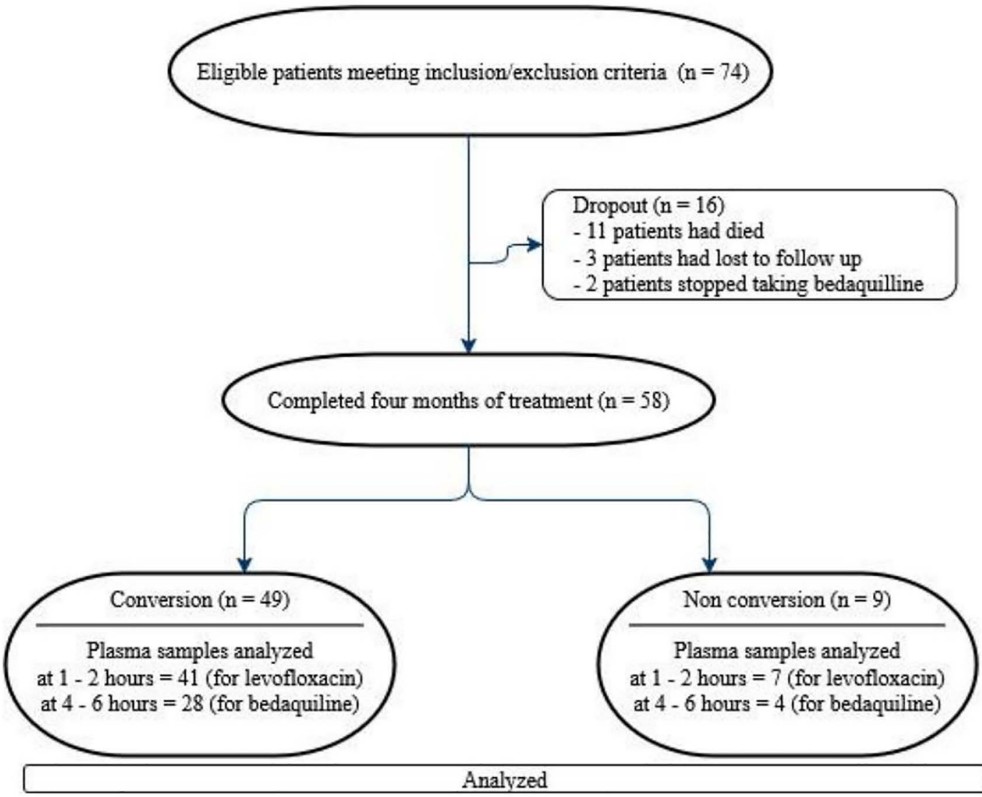

**Fig 1. Patient recruitment and follow-up flowchart.**

[23,24]. Fig 5 shows the levofloxacin Cmax/MIC ratio, which refers to the target Cmax/MIC ratio for antibiotics that show a concentration-dependent activity pattern [24].

Sixteen patients who dropped out of the study consisted of patients who were lost to follow-up (three patients), died (11 patients), and discontinued bedaquiline due to the QTc prolongation (two patients). Those patients who were lost to follow-up had experienced nausea and vomiting, edema in the ankles and feet, and, for some, the problem of a long distance from home to the hospital. The cause of death for eleven patients was septic shock, pneumonia, and cardiac arrest (seven patients), while the cause of death of the other four patients at home was unknown.

## Discussion

Our study showed that the Cmax/MIC ratio of bedaquiline was significantly lower in the non-conversion group compared to the conversion group. In contrast, there was no significant difference between the Cmax/MIC ratio of levofloxacin between the two groups.

Bedaquiline Cmax was significantly different between the sputum conversion group and the non-conversion group (1.54 mg/L vs. 0.70 mg/L; p = 0.041), which is consistent with the findings of the previous study [25]. Shao et al showed that the data from 159 patients with MDR-TB, the bedaquiline Cmax during the early phase of MDR-TB treatment was generally higher in patients with sputum conversion, as well as in patients with a successful treatment outcome at the end of treatment [25]. Higher bedaquiline exposure, as part of a treatment regimen, had a beneficial effect on clinical outcomes [26]. The mean Cmax/MIC ratio of bedaquiline was higher in the sputum conversion group compared with the non-conversion group (9.10 vs. 1.65; p < 0.001). However, 26 patients who had Cmax/MIC of bedaquiline in the conversion

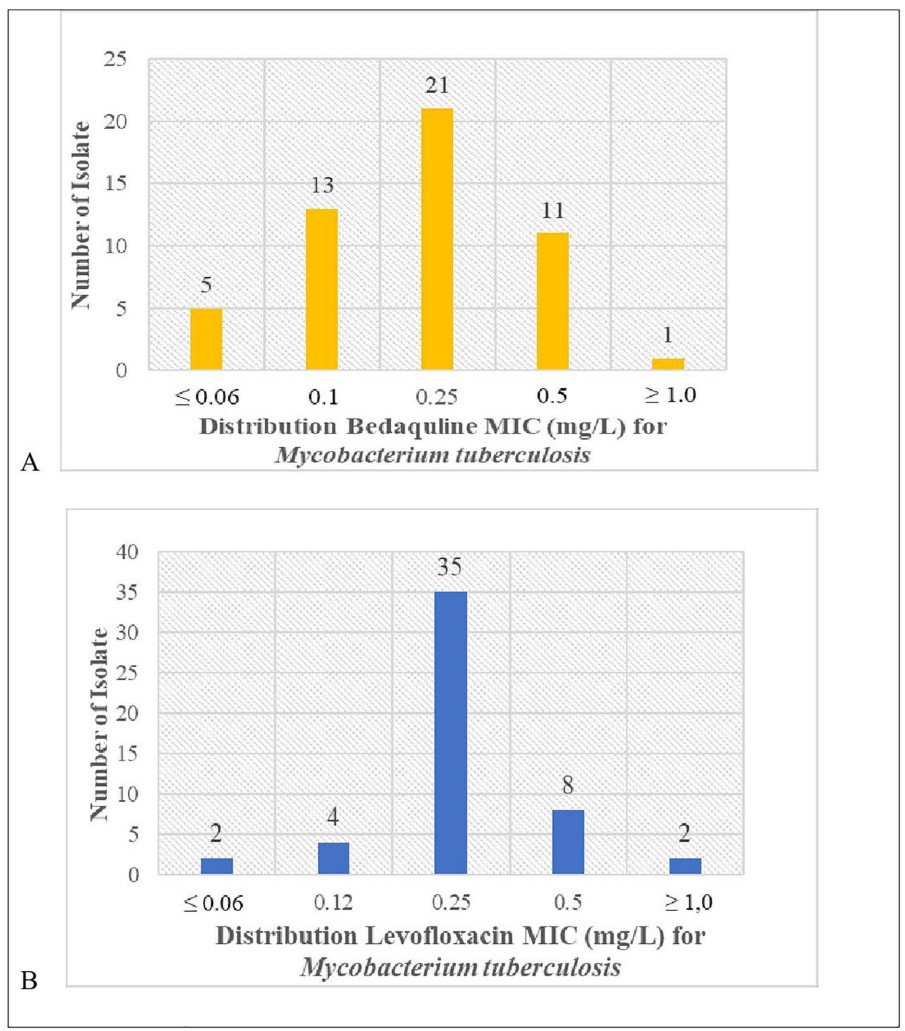

**Fig 2. (A) Minimum inhibitory concentration distribution of bedaquiline and (B) Minimum inhibitory distribution of levofloxacin in 51 resistant *Mycobacterium tuberculosis* isolates performed in *Mycobacteria* Growth Indicator Tube (MGIT).**

**Table 2. Comparison of bedaquiline MIC and levofloxacin MIC with sputum conversion (n = 51).**

| Variables | Conversion | Non-conversion | P value* |
|---|---|---|---|
| Bedaquiline MIC, n (%) | | | |
| <0.25 | 14 (41.9) | 2 (28.0) | 0.315 |
| ≥0.25 | 25 (58.1) | 6 (75.0) | |
| Levofloxacin MIC, n (%) | | | |
| <0.25 | 5 (11.6) | 1 (12.5) | 0.661 |
| ≥0.25 | 38 (88.4) | 7 (87.5) | |

Notes: MIC, Minimum Inhibitory Concentration (MICs of bedaquiline and levofloxacin against Mtb); *Fisher's exact test

**Table 3. Comparison of levofloxacin Cmax/MIC ratio in MDR-TB patients with sputum conversion.**

| Levofloxacin variables | Conversion | Non-conversion | p value |
|---|---|---|---|
| | n=41 | n=7 | |
| Cmax in mg/L***, mean (SD) | 11.36 (5.69) | 10.97 (6.56) | 0.873* |
| | n=36 | n=6 | |
| Cmax/MIC ratio, median (min-max) | 45.64 | 41.72 | 0.941** |
| | (17.14–206.23) | (23.06–86.68) | |

*unpaired t test; **Mann-Whitney U test; significant if the p value <0.05; ***1–2 hours after drugs administration

**Table 4. Comparison of bedaquiline Cmax/MIC ratio in MDR-TB patients with sputum conversion.**

| Bedaquiline variables | Conversion | Non-conversion | p value* |
|---|---|---|---|
| | n=28 | n=4 | |
| Cmax in mg/L**, mean (SD) | 1.54 (0.76) | 0.70 (0.44) | **0.041** |
| | n=26 | n=4 | |
| Cmax/MIC ratio, mean (SD) | 9.10 (7.21) | 1.65 (1.58) | **< 0.001** |

*unpaired t test; significant if the p value <0.05; **4–6 hours after drugs administration.

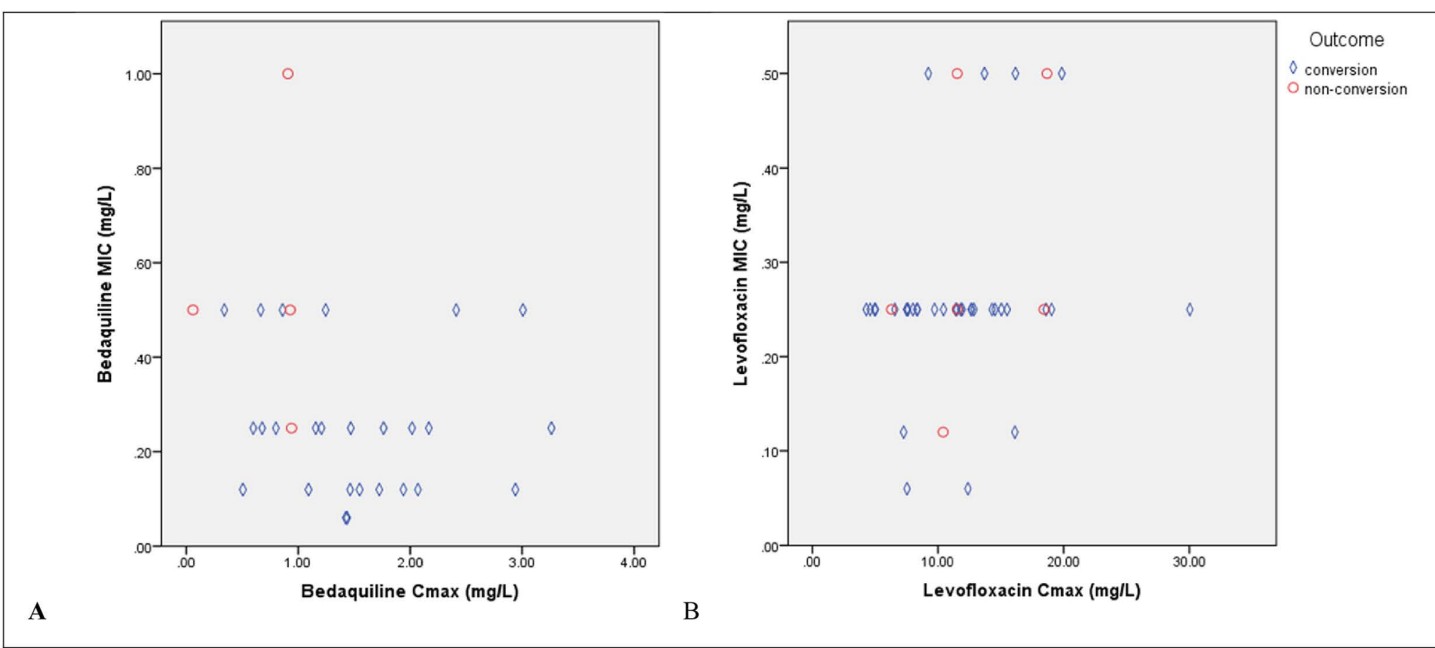

**Fig 3. Summary of the outcome (conversion or non-conversion) based on levofloxacin, bedaquiline Cmax, and their minimum inhibitory concentrations (MICs) (A and B).**

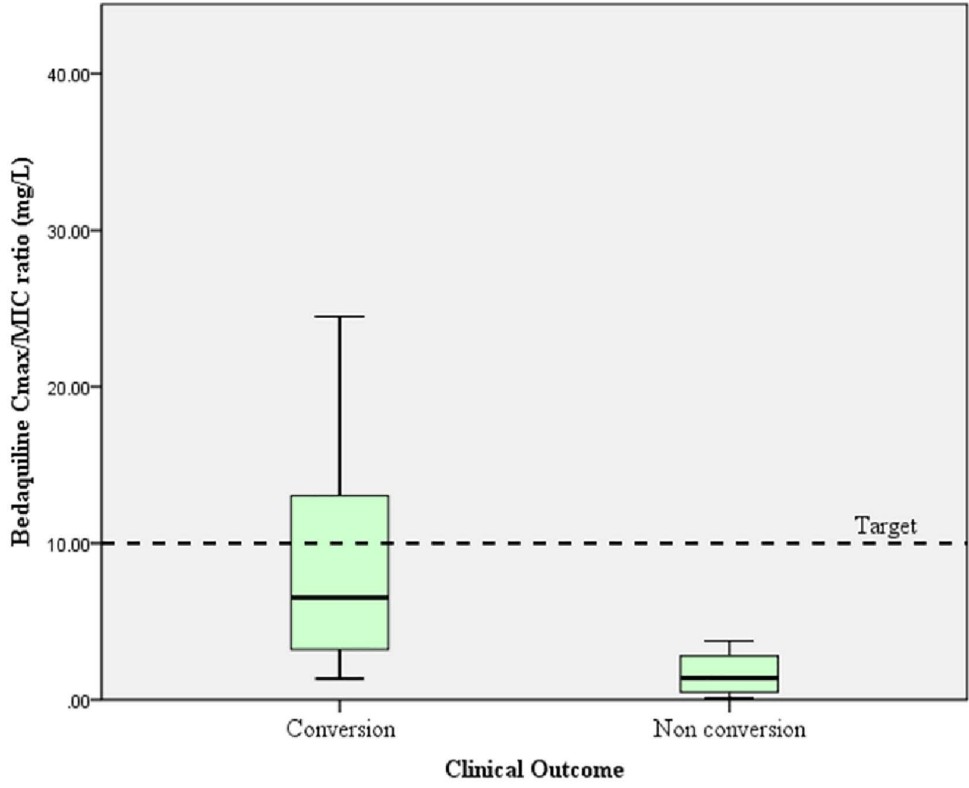

**Fig 4. Comparison of bedaquiline Cmax/MIC ratio in MDR-TB patients with sputum conversion and the targeted ratio.** ▬ ▬ ▬ ▬
Note. Targeted Cmax/MIC ratio > 10 (The targeted Cmax/MIC ratio > 10 is a concentration-dependent activity of antibiotics*). *Azanza Perea JR. The pharmacodynamic bases of the prescription of antimicrobials. Rev Esp Quim. 2019;(2):35–7.

groups and four patients in the non-conversion groups (26 vs. 4) resulted in an imbalance in the number of patients in different condition groups within a study. This imbalance can be caused by consecutive patient enrollment or reflects the real-world distribution of cases during the cohort study period. In general, the baseline characteristics of patients showed no significant difference between groups [27].

The magnitude of the MIC value and the pharmacokinetic/pharmacodynamic parameter of bedaquiline will impact the clinical outcomes [28]. In this study, we evaluated whether patients infected with Mtb strains that had higher MIC values might have different outcomes compared to patients infected with Mtb strains that had lower MIC values. Meanwhile, there was no significant difference between the group with MIC < 0.25 mg/L and the group with MIC ≥ 0.25 mg/L in comparison to sputum conversion. The existing MIC grouping may not provide more information than the pharmacokinetic/pharmacodynamic (Cmax/MIC) approach, because this approach encompasses interactions between the host, pathogen, and drug. [29,30].

A study in China showed that the bedaquiline MIC was 0.06 mg/L, which is lower than ours. Hence, bedaquiline Cmax/MIC for both conversion and non-conversion groups (41.1 vs. 6.2) was higher than in our study [26]. Differences in bedaquiline MIC values between populations may indicate that there are varying levels of Mtb susceptibility to bedaquiline. It is not known yet whether it is influenced by mutations in genes associated with bedaquiline targets, such as the atpE, Rv0678, Rv2535, and Rv1979 mutations. Our study did not study gene mutations [12,31].

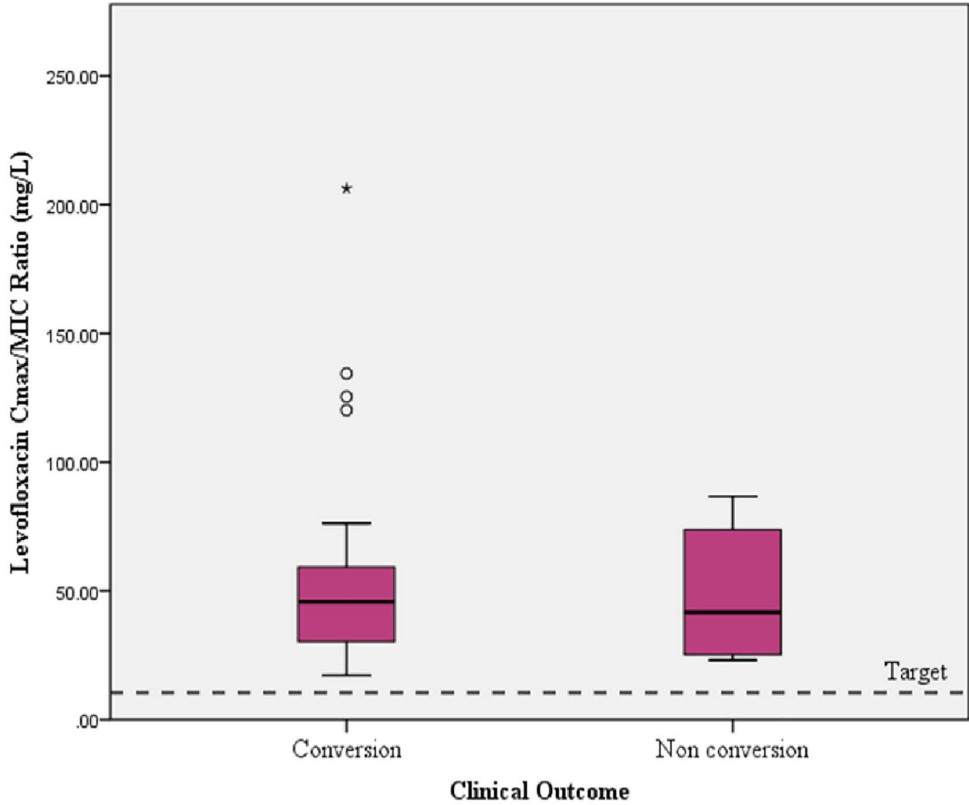

**Fig 5. Comparison of levofloxacin Cmax/MIC ratio in MDR-TB patients with sputum conversion and the targeted ratio.** ━ ━ ━ ━

 Note. Targeted Cmax/MIC ratio > 10 (the targeted Cmax/MIC ratio > 10 is a concentration-dependent activity of antibiotics*). *Azanza Perea JR. The pharmacodynamic bases of the prescription of antimicrobials. Rev Esp Quim. 2019;(2):35–7.

However, based on previous studies, the ideal Cmax/MIC threshold should be greater than 10 for a concentration-dependent activity of antibiotics [24,32,33]. Four out of 30 patients did not achieve sputum conversion within four months of treatment. In particular, their bedaquiline MIC was 0.5 mg/L (three patients) and 1 mg/L (one patient), thus resulting in a low Cmax/MIC ratio. The possible reason is that patients who have had a previous TB treatment can lead to increased MIC values [34]. Furthermore, administration of medicines without food may contribute to the lower plasma bedaquiline concentration, mean 0.70 (SD 0.44) mg/L, as a high-fat diet markedly increases the speed and extent of gastrointestinal absorption of bedaquiline [35]. In this study, the mean bedaquiline Cmax of the non-conversion group was 0.70 (SD 0.44) mg/L. In addition, covariate factors such as body weight, albumin level, and race were associated with bedaquiline plasma concentration. Body weight and albumin level were typically correlated in MDR TB patients. Bedaquiline is highly protein-bound (> 99.9%); low albumin levels are expected to impact the bedaquiline clearance and volume distribution [26,36]. Plasma concentration monitoring may help to assess whether the patient is achieving adequate drug concentrations for sputum conversion, and there is a need to identify potential drug absorption or metabolism issues that may cause delayed sputum conversion [21,36,37].

In this study, the mean of levofloxacin Cmax was within the range of expected Cmax, in both conversion and non-conversion groups, which were 11.36 and 10.97 mg/L, respectively. From several pharmacokinetic studies, the expected Cmax after a standard dose (700 or 1000 mg) of levofloxacin based on body weight was 8–13 mg/L [13,25]. There was

no significant difference in levofloxacin Cmax between the sputum conversion and non-conversion groups (p = 0.873). According to the MIC of levofloxacin, most of the MICs of levofloxacin against Mtb in this study were 0.25 mg/L (68.6%), except for two MICs, which were 1 mg/L. These MIC values indicated low variability in the distribution of levofloxacin MIC against Mtb. We found that the median of the Cmax/MIC ratio of levofloxacin appeared to be higher than the targeted Cmax/MIC ratio in the conversion and non-conversion groups (43.68 and 41.72, respectively), and there was no significant difference between the two groups (p = 0.987). The levofloxacin Cmax/MIC ratio was greater than 10, including six patients (14%) in the non-conversion group. This finding was in contrast to the bedaquiline Cmax/MIC ratio, which may indicate several potential factors. First, the high levofloxacin Cmax/MIC ratio alone may not significantly suggest the early sputum conversion, as the variability in the distribution of levofloxacin MIC may be lower than that of bedaquiline MIC. Second, the Cmax of levofloxacin was not different according to the clinical outcome. Third, the limitation in the number of study samples for the levofloxacin Cmax/MIC variable may not capture the clinical variability; therefore, a larger number of patients is needed to validate these findings. Fourth, the area under the curve (AUC) of plasma concentration and the MIC might be a more relevant parameter to further ensure the clinical outcome. However, AUC plasma concentration measurement requires more than one point sampling, which cannot be performed in this study due to technical limitations [38,39]. Fifth, there was a delay in sputum conversion caused by inadequate exposure of other anti-TB drugs, such as bedaquiline Cmax, in the regimen. Delayed sputum conversion may also be due to individual factors such as low BMI (< 18.5 kg/m2), which is usually associated with malnutrition accompanied by low levels of immunoglobulins, interleukin-2 receptors, and various T cell subsets [40,41]. Among patients in the non-conversion group, some individuals had median MCV levels of 79 (71–84) fL, which could be considered as microcytic anemia [42,43]. Microcytic anemia may reflect an underlying nutritional status related to chronic disease [43,44]. Meanwhile, four out of nine patients in the non-conversion group received iron supplements. Earlier studies found interactions between iron supplements and levofloxacin, which in these cases may cause a decrease in the absorption and effectiveness of levofloxacin [45–47]. However, in this study, we did not study the possibility of interaction. In long-term co-administration of levofloxacin and other anti-TB drugs, such as bedaquiline and clofazimine, further study is needed to investigate the impact of drug interactions on long-term safety, such as QT interval prolongation. [48].

This study has several limitations. First, this study had some missing Cmax data for bedaquiline or levofloxacin due to technical limitations; however, there were no significant differences between patients with missing data and those without missing data. Second, there were no serial albumin levels during treatment. Third, the study was not designed to evaluate drug-drug interaction in TB treatment; we only studied two core drugs. Despite these limitations, our research provides the first information on plasma concentrations of bedaquiline and levofloxacin, as well as their MICs in Indonesian patients with MDR-TB.

In summary, our study found higher Cmax/MIC of bedaquiline in the conversion group compared with the non-conversion group within four months during the early phase of MDR-TB treatment in Indonesia. This funding suggests a potential relationship between the bedaquiline Cmax/MIC ratio and sputum conversion in patients with MDR-TB. A similar relationship study should also be done for levofloxacin, as in this study, the number of samples for the levofloxacin Cmax/MIC variable was insufficient to conclude. It may be useful to have MIC data for main TB drugs in different countries and to assess the bedaquiline Cmax/MIC ratio in specific individual cases during the early phase of MDR-TB treatment. However, the possibility of implementation of this plasma concentration monitoring in clinical practice should be carefully considered and supported by further research in various settings.

## Supporting information

**S1 File. Sample size calculation output (screenshots and MedCalc parameters).**
(DOCX)

**S2 Table. Demographic data and clinical characteristics of 58 MDR-TB patients at the start of treatment (baseline) by whether bedaquiline or levofloxacin Cmax is missing or not.**
(DOCX)

**S3 Table. Summary of assessing normality test using Shapiro-Wilk, skewness, and kurtosis for a characteristic of variables in various samples.**
(XLSX)

**S4 Table. Mean, confidence interval, median, variance of bedaquilin and levofloxacin Cmax, Cmax/MIC ratio in MDR TB patients with sputum conversion and non-conversion groups.**
(XLSX)

**S5 Table. Additional baseline characteristics of 30 patients who had Cmax/MIC bedaquiline data.**
(DOCX)

## Acknowledgments

We would like to thank the hospital directors, dr Ida Bagus Sila Wiweka, Sp.P (K), MARS and dr Monika Saraswati Sitepu, M.Sc, the doctors, the nurses, and the laboratory staff of the National Infectious Disease Center Sulianti Saroso Hospital, Pasar Rebo Regional General Hospital and Pulmonary Hospital Dr. M. Goenawan Patoidigdo for their support. This study was also supported by the Indonesian Endowment Fund for Education (LPDP) on behalf of the Ministry of Higher Education, Science, and Technology of the Republic Indonesia, under the EQUITY Program (contract No. 4302/BT/DT.03.08/2025 and 573/PKS/R/UI/2025).

## Author contributions

**Conceptualization:** Nina Mariana, Anggi Gayatri, Indah Suci Widyahening, Budiman Bela, Soedarsono Soedarsono, Erni Juwita Nelwan, Purwantyastuti Ascobat.

**Data curation:** Nina Mariana, Iin Maemunah, Vivi Setiawaty, Erni Juwita Nelwan.

**Formal analysis:** Nina Mariana, Anggi Gayatri, Indah Suci Widyahening, Erni Juwita Nelwan.

**Funding acquisition:** Nina Mariana, Anggi Gayatri, Vivi Setiawaty, Vithal Prasad Myneedu, Purwantyastuti Ascobat.

**Investigation:** Nina Mariana.

**Methodology:** Nina Mariana, Indah Suci Widyahening, Purwantyastuti Ascobat.

**Project administration:** Iin Maemunah, Vivi Setiawaty, Vithal Prasad Myneedu, Erni Juwita Nelwan, Purwantyastuti Ascobat.

**Resources:** Iin Maemunah.

**Supervision:** Anggi Gayatri, Indah Suci Widyahening, Budiman Bela, Soedarsono Soedarsono, Erni Juwita Nelwan, Purwantyastuti Ascobat.

**Validation:** Nina Mariana, Anggi Gayatri, Indah Suci Widyahening, Budiman Bela, Soedarsono Soedarsono, Erni Juwita Nelwan.

**Visualization:** Purwantyastuti Ascobat.

**Writing – original draft:** Nina Mariana, Purwantyastuti Ascobat.

**Writing – review & editing:** Anggi Gayatri, Indah Suci Widyahening, Soedarsono Soedarsono, Vithal Prasad Myneedu, Erni Juwita Nelwan, Purwantyastuti Ascobat.

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
