## [Decision Letter · Decision Letter 0]

12 Aug 2025

Dear Dr. Mariana,

Thank you for submitting your manuscript to PLOS ONE. After careful consideration, we feel that it has merit but does not fully meet PLOS ONE’s publication criteria as it currently stands. Therefore, we invite you to submit a revised version of the manuscript that addresses the points raised during the review process.

We look forward to receiving your revised manuscript.

Kind regards,

Saki Raheem, PhD

Academic Editor

PLOS ONE

Journal Requirements:

[Universitas Indonesia   http://dx.doi.org/10.13039/501100006378    NKB_292   Prof Purwanty Ascobat]. 

4. Thank you for stating the following in your manuscript:

[This study was funded by the Directorate of Research and Development, Universitas Indonesia, under Hibah PUTI 2024 (Grant No NKB_292).]

[The author(s) received no specific funding for this work.]

5. In the online submission form, you indicated that [Data cannot be shared publicly due to restrictions. Data are available from the Institutional Data Access / Ethics Committee (contact via first author) for researchers who meet the criteria for accessing confidential data].

7. Please amend the manuscript submission data (via Edit Submission) to include author Purwantyastuti Ascobat

8. Please amend your authorship list in your manuscript file to include author Purwanty Ascobat.

Reviewers' comments:

Reviewer's Responses to Questions

**Comments to the Author**

1. Is the manuscript technically sound, and do the data support the conclusions?

Reviewer #1: Yes

Reviewer #2: Partly

2. Has the statistical analysis been performed appropriately and rigorously?

Reviewer #1: Yes

Reviewer #2: No

3. Have the authors made all data underlying the findings in their manuscript fully available?

Reviewer #1: Yes

Reviewer #2: No

4. Is the manuscript presented in an intelligible fashion and written in standard English?

Reviewer #1: Yes

Reviewer #2: No

Reviewer #1: This study addresses an important public health issue in a high TB-burden setting by evaluating pharmacokinetic/pharmacodynamic (PK/PD) parameters, specifically the Cmax/MIC ratio of bedaquiline and levofloxacin, and their association with early sputum conversion in MDR-TB patients.

The research is relevant and executed within clinical contexts reflective of Indonesian practice.

However, several aspects require critical clarifications.

The number of evaluable patients for bedaquiline (n = 32) and levofloxacin (n = 48) Cmax measurements is relatively small and unbalanced between conversion and non-conversion groups, especially for bedaquiline (only 4 non-converters).

While the authors report a significant difference in bedaquiline Cmax/MIC between conversion and non-conversion groups, the small number of non-converters (n = 4) undermines confidence in the statistical validity of this finding. The assumption of normality for parametric tests (e.g., t-test) with such small group sizes is questionable.

The lack of significance in levofloxacin Cmax/MIC differences is presented without adequate discussion.

Reviewer #2: The authors looked at the extent to which the peak dosage at 1 month of 2 antibiotics given for multidrug-resistant tuberculosis (levofloxacinand bedaquiline) was associated with microbiological success (sputum negativation at 4 months).

This is a relevant objective, and the data are interesting.

However, we are hampered by the absence of an essential piece of data: the minimum inhibitory concentrations for the 2 antibiotics according to the outcome (sputum conversion or not). MIC are described in general, but not per outcome.

The authors observe that the ratio between the maximal concentration and the MIC does indeed differ according to microbiological success, However, the variable most strongly associated with this success may be the MIC rather than the ratio Cmax/MIC.

The fact that the ratio peak/MIC is not different according to the microbiological sucess for levofloxacin is maybe due to the facts that 1) MIC for levofloxacin are for all but 2 cases under the breakpoint (1 mg/L), and 2) the concentration are not different according to outcome (11.4 vs 11.0 mg/L).

Meanwhile, for bedaquiline, most of the strains are above the "clinical breakpoint" (see https://www.who.int/publications/i/item/WHO-CDS-TB-2018.5). Therefore, when the authors report the important difference in Cmaw according to the outcome, it would be far more clear if they also provide MIC. For example, a figure showing in the X axis the Cmax and in the Y axis the MIC, each case represented by a point, and with different point shape or color according to the outcome, may add clarity.

Moreover, the authors perform a peak concentration for levofloxacin as if it was a legitimate parameter for an antibiotic which they describe as dose-dependent, but, for fluoroquinolones, the parameter most often reported as relevant is the ratio between the air under the curve (AUC) and the MIC, and measuring AUC requires more than just one point.

In addition, the author report that 58 patients are analysed; meanwhile, they have data for Cmax and MIC for only 48 patients for levofloxacin, and 32 for bedaquiline. Why announce 58 in the abstract?

In the same line, it is not appropriate to perform statistical tests with numbers so small (ex: non-conversion group: only 4 Cmax values for bedaquiline).

At last, multiple sentences should be reviewed, being structured in a strange way or lacking verb.

**Do you want your identity to be public for this peer review?** For information about this choice, including consent withdrawal, please see our Privacy Policy

Reviewer #1: No

Reviewer #2: No

---

## [Author Response · Author response to Decision Letter 1]

29 Aug 2025

We would like to thank editor and reviewers for all the comments and suggestions. We have carefully revised the manuscript in response to the reviewers’ feedback. We are enclosing herewith the revised manuscript entitled “The Peak Plasma Concentration (Cmax)/Minimum Inhibitory Concentration (MIC) of Bedaquiline and Levofloxacin with Special Attention to the Sputum Conversion in the Treatment of Multidrug-Resistant Tuberculosis in Indonesia.” A point-by-point response to all reviewers’ comments has been provided in the attached document.

---

## [Decision Letter · Decision Letter 1]

15 Sep 2025

Dear Dr. Mariana,

Thank you for submitting your manuscript to PLOS ONE. After careful consideration, we feel that it has merit but does not fully meet PLOS ONE’s publication criteria as it currently stands. Therefore, we invite you to submit a revised version of the manuscript that addresses the points raised during the review process.

We look forward to receiving your revised manuscript.

Kind regards,

Saki Raheem, PhD

Academic Editor

PLOS ONE

Reviewers' comments:

Reviewer's Responses to Questions

**Comments to the Author**

Reviewer #1: All comments have been addressed

Reviewer #2: (No Response)

2. Is the manuscript technically sound, and do the data support the conclusions?

Reviewer #1: Yes

Reviewer #2: Yes

3. Has the statistical analysis been performed appropriately and rigorously?

Reviewer #1: Yes

Reviewer #2: No

4. Have the authors made all data underlying the findings in their manuscript fully available?

Reviewer #1: Yes

Reviewer #2: Yes

5. Is the manuscript presented in an intelligible fashion and written in standard English?

Reviewer #1: Yes

Reviewer #2: No

Reviewer #1: The number of samples were increased in supplement section in all conditions. Also, 3. the lack of significance in levofloxacin Cmax/MIC differences is now presented in discussion. Thus, all concerns are well addressed in the revised version.

Reviewer #2: The authors have addressed most of the comments from the reviewers.

This is important, as the data reported deserve publication.

There is still some issues with some points:

for the abstract:

- "58 patients completed the analysis at four months of treatment.": again, no. Your article title is on antibiotic concentration, so you cannot say that 58 patients completed the study if 10 of them did not. The sentence that corrects this figure just after is a point, but you should straightforward say 48, not 58.

- "There was a statistically significant difference between the Cmax/MIC ratio of bedaquiline in the sputum conversion and non-conversion groups (9.10 vs. 1.65; p < 0.001).": again, you cannot do a statistical analysis with 3 or 4 patients in one of the group.

for the text:

- "The MIC values for bedaquiline and levofloxacin, based on clinical outcome (sputum conversion or non-conversion), did not show significant differences (see S3 Table).": I is a good thing that the authors added this data. However, this is a major data, it should not be in a supplementary table, but as a part of the main table describing the particularities of the patients included.

- "In this study, MIC bedaquiline 0.25 mg/L was 21 (41.2%) Mtb isolates,": this fragment sentence makes no sense, a word or a mathematical sign is lacking.

- "Meanwhile, MIC alone could not be compared with the clinical outcome (sputum conversion)": this sentence is not clear. What would one want to compare a MIC with non-MIC data? Maybe the authors want to say " Groups made according to MIC alone could not be compared with the clinical outcome (sputum conversion)"; but if it is what they had in mind, I do not understand why - it is basic practice to see whether people infected with a bacterial strain with high MIC have or not an outcome different from people infected wiht a bacterial strain with lower MIC.

- "MIC distribution of bedaquiline for Mtb was presented in this study, which showed high pharmacodynamic variability.": it is a bit confusing to say so, as pharmacodynamic also result from PK. Moreover, there is not "pharmacodynamic variability"; conversely, pharmacodynamic analyses lead to variable conclusions according to PK and to MIC

- "According to the MIC of levofloxacin, most of the MICs of levofloxacin against Mtb in this study were 0.25 mg/L (68.6%)": the sentence makes no sense; maybe the authors want to say "Regarding the MIC of levofloxacin, most (68.6%) of the Mtb strains in this study were under the value of 0.25 mg/L" (or "were equal to 0.25 mg/L").

for the new figure 3: remove the lines of "correlation": the idea was not to see a correlation (this would be completely nonsense in this case), but to better visualise the data. (Moreover, it is not possible to draw a correlation line with 3 dots...)

**Do you want your identity to be public for this peer review?** For information about this choice, including consent withdrawal, please see our Privacy Policy

Reviewer #1: No

Reviewer #2: No

---

## [Author Response · Author response to Decision Letter 2]

1 Oct 2025

We thank the reviewer and editor for the continued feedback. We have revised the manuscript point-by-point in the responses to reviewer file. We hope this revision file addresses the reviewers' concerns.

---

## [Decision Letter · Decision Letter 2]

23 Oct 2025

The Peak Plasma Concentration (Cmax)/Minimum Inhibitory Concentration (MIC) of Bedaquiline and Levofloxacin with Special Attention to the Sputum Conversion in the Treatment of Multidrug-Resistant Tuberculosis in Indonesia

PONE-D-25-28090R2

Dear Dr. Mariana,

We’re pleased to inform you that your manuscript has been judged scientifically suitable for publication and will be formally accepted for publication once it meets all outstanding technical requirements.

Kind regards,

Saki Raheem, PhD

Academic Editor

PLOS ONE

Reviewers' comments:

Reviewer's Responses to Questions

**Comments to the Author**

Reviewer #2: All comments have been addressed

2. Is the manuscript technically sound, and do the data support the conclusions?

Reviewer #2: (No Response)

3. Has the statistical analysis been performed appropriately and rigorously?

Reviewer #2: (No Response)

4. Have the authors made all data underlying the findings in their manuscript fully available?

Reviewer #2: (No Response)

5. Is the manuscript presented in an intelligible fashion and written in standard English?

Reviewer #2: (No Response)

Reviewer #2: (No Response)

**Do you want your identity to be public for this peer review?** For information about this choice, including consent withdrawal, please see our Privacy Policy

Reviewer #2: **Yes: ** Olivier Epaulard

---

## [Editor Report · Acceptance letter]

PONE-D-25-28090R2

PLOS ONE

Dear Dr. Mariana,

I'm pleased to inform you that your manuscript has been deemed suitable for publication in PLOS ONE. Congratulations! Your manuscript is now being handed over to our production team.

Kind regards,

on behalf of

Dr. Saki Raheem

Academic Editor

PLOS ONE